

# Coriolis Recovery of Wind Farm Wakes

Ronald B. Smith[1] and Brian J. Gribben[2]

[1]Department of Earth and Planetary Sciences, Yale University, New Haven, CT 06520, USA
[2]Frazer-Nash Consultancy Ltd., Bristol, BS1 4BA, UK

*Correspondence to*: Brian J. Gribben (b.gribben@fnc.co.uk)

**Abstract.** Two mechanisms cause wind speed recovery in the wake of a wind farm: momentum mixing and the Coriolis force. To study these mechanisms, we use a steady linearized two-layer Fast Fourier Transform (FFT) model so that both analytical expressions and full flow fields can be derived. The model parametrizes the vertical momentum mixing as Rayleigh friction. Pressure gradient forces are computed using a two-part vertical wave number formulation in the upper layer.  The Coriolis

force recovery occurs by deflecting flow leftward (in the northern hemisphere). The Coriolis force, acting on this crossflow, re-accelerates the flow in the downwind direction.

The relative importance of Rayleigh versus Coriolis wake recovery depends on their two coefficients: $C$ and $f$ respectively, each with units of inverse time. When the coefficient ratio is large, i.e. $\dfrac{C}{f} >> 1$ , Rayleigh friction restores the wake before

Coriolis can act. Farm size and atmospheric static stability are also important to wake recovery. The wakes of small and medium size farms will quickly approach geostrophic balance.  Once balance is established, the ratio of farm size "$a$" to the Rossby Radius of Deformation (RRD) determines the amount of Coriolis recovery.  For a small farm in a stable atmosphere ($a <$ RRD), Coriolis acts by adjusting the pressure field to obtain geostrophic balance rather than accelerating the wind.  When this occurs, only momentum mixing can restore the "inner" wake. For large farms in less stable conditions ($a >$ RRD), the

Coriolis Force significantly contributes to wake recovery. In this case, the leftward deflected flow creates "edge jets" on either side of the wake.  Including the Coriolis force when modelling wind farm wakes is demonstrated to be increasingly important for larger wind farms or farm clusters.

## 1 Introduction

We investigate the role of the Coriolis force on wind farm wake recovery.  Wake recovery has a large literature, but mostly

focused on the role of turbulence in restoring the flow by mixing momentum from the ambient airstream back into the wake, both laterally and vertically.  See reviews by Stevens and Meneveau (2017), Archer et al. (2018), Porté-Agel et al. (2020), Pryor et al. (2020), and Fischereit et al. (2021).





Another potential recovery mechanism is the generation of gravity waves (Smith 2010, 2022, 2024; Allaerts and Meyers 2017,
2019; Devesse et al. 2022, Khan et al. 2024). Our current interpretation of this previous work is that  the pressure gradients
from gravity waves  act mostly locally with little impact on the far field wake recovery.

The literature on a Coriolis force recovery mechanism is now growing. Van der Laan and Sørensen (2017) used a Reynolds-
Averaged Navier-Stokes (RANS) numerical model to see how two medium-size wind farms influence each other. Their model
includes the role of wind veering with height in the regional boundary layer as well as local farm-induced pressure and Coriolis
forces. They found that the wake slightly turned to the right under the influence of the entrained veered momentum.  Earlier
work from van der Laan et al. (2015), also using a RANS model, noted that the expected left turn when the flow is decelerated
(at the turbine) is not visible, and the right turn as it is accelerated (in the wake recovery zone) dominates because there is more
time and a greater length scale for the deflection to take effect. Gadde and Stevens (2019) used a large eddy simulation (LES)
model with veering wind and confirmed the rightward turning. Nygaard and Newcombe (2018) found evidence of it in Doppler
radar data.  Narasimhan et al. (2024) developed a quasi-analytic model of wakes in a veering boundary layer. These papers
have not examined inertial wave generation and geostrophic adjustment. A broader look at Coriolis effects was given in Maas
(2023). That paper used a full physics LES model to compare a 13 km and 90 km long wind farm of infinite width and found
significant differences, also observing turning to the left within the farm and turning to the right in the wake.  A similar full-
physics approach was used by Heck and Howland (2024) to look at Coriolis effects on individual turbine wakes.  None of
these previous analyses have examined the role of static stability and geostrophic adjustment in the wake.

In Section 2, we review the classical idea of geostrophic adjustment in a stratified fluid on a rotating planet that provides a
foundation for this paper.   In Section 3, we formulate a linearized steady-state two-layer problem with turbine drag applied to
the lower layer. In Section 4, we find an idealized but instructive 1-D flow field solution for damped inertial waves. In Section
5 we find more useful 3-D solutions using Fourier Transforms. Using these solutions, we analyse the global competition
between Coriolis and Rayleigh forces to recover the wake.  In Section 6, we display wake solutions from Fast Fourier
Transform (FFT) calculations. In Section 7, we describe the forces on air parcels passing though the wind farm. In Section 8,
we explain how the wake approaches geostrophic balance in a stable atmosphere. In Section 9, we discuss wind power
55    applications of the new theory.

## 2 Geostrophic Adjustment

The concept of geostrophic balance and the process of geostrophic adjustment are important in atmospheric and ocean
dynamics (Rossby 1938, Blumen 1972, Lewis 1996, Chagnon and Bannon 2005, Mak 2011). We review these ideas here as a
foundation for our wake recovery analysis. In the classical shallow layer adjustment problem, a band of air is suddenly put
60    into motion with no surface tilt or pressure gradient. The Coriolis force pushes the band to the right (in the northern





hemisphere). This rightward shift does two things. First, it generates a Coriolis force that slows the band and second, it piles up air to the right and evacuates the left side creating a cross-flow pressure gradient. Together, these two processes restore geostrophic balance after an elapsed time of about $T = 1/f$ where $f$ is the Coriolis parameter. In mid-latitudes $f \approx 0.0001\ s^{-1}$ so $T$ is about 3 hours. A key aspect of geostrophic adjustment is the role of the Rossby Radius of Deformation (RRD). When the band of accelerated wind is wider than the RRD, the geostrophic adjustment occurs mainly by altering the wind speed. However, when the band width is less than the RRD, the adjustment occurs by creating a balancing pressure gradient rather than recovering the wind speed.

The steady wind farm problem considered here is similar to Rossby's classic problem but instead of a temporal evolution, an upwind balanced flow is locally slowed by wind farm drag and eventually returns to geostrophic balance downwind. Thus, the adjustment occurs in space, not in time. Like Rossby, we adopt a two-layer formulation with a uniform lower layer and stratification aloft. Our analysis of geostrophic adjustment in the wind farm context includes frictional dissipation as well as gravity wave and inertial wave generation. The related problem of a secondary circulation caused by frictional retardation was discussed by Eliassen (1952) and Egger (2003).

## 3 Turbine Layer Formulation

### 3.1 Governing equations

The airflow in the lower "turbine layer" can be analysed using these linearized steady perturbation momentum equations.

$$U\frac{\partial u}{\partial x} + V\frac{\partial u}{\partial y} = F_x - \left(\frac{1}{\rho}\right)\frac{\partial p}{\partial x} - Cu + K\nabla^2 u + fv \tag{1a}$$

$$U\frac{\partial v}{\partial x} + V\frac{\partial v}{\partial y} = F_y - \left(\frac{1}{\rho}\right)\frac{\partial p}{\partial y} - Cv + K\nabla^2 v - fu \tag{1b}$$

where $F_x(x,y)$, $F_y(x,y)$ are the two components of the turbine drag with units of acceleration. The second term on the right is the pressure gradient force (PGF). Symbols $C$ and $K$ are the coefficients of Rayleigh friction and lateral momentum diffusivity. This formulation is consistent with that used in previous work (Smith 2010), and here also includes the lateral momentum diffusion term and Coriolis term as has been used by other authors (e.g. Allaerts and Meyers 2019). The derivation of these equations, the depth-averaging approach, and the linearization procedure are well established in the literature so further detail is omitted here. The components of undisturbed, depth averaged wind speed in the horizontal $x, y$ plane are represented by $U, V$ respectively. The corresponding perturbation wind speeds and pressure are $u(x, y), v(x, y)$ and $p(x, y)$. The air density $\rho$ here is a constant. The formulation used allows for different wind speeds and directions in the lower, turbine layer and in the upper layer, however in this work we assume the same wind speed and direction in both layers.



## 3.2 Coriolis force

The Coriolis force in Eq. (1) is a deflecting force acting on objects moving horizontally on our rotating planet. This force is proportional to the Coriolis parameter $f$ where

$$f = 2\Omega \, \sin(\phi) \tag{2}$$


Here $\phi$ is latitude and the rotation rate of the earth is $\Omega \approx 7.29 \times 10^{-5}$ radians per second. The signs of the $f$ terms in Eq. (1) ensure that the Coriolis force acts perpendicularly to the velocity vector. We assume the background flow $\vec{U} = (U, V)$ and pressure field $P(x, y)$ are in geostrophic balance


$$\frac{\nabla P}{\rho} = -f \, \vec{k} \times \vec{U} \tag{3}$$

where $\vec{k}$ is the unit vector in the vertical direction. Any velocity perturbation $\vec{u} = (u, v)$ will cause a perturbation Coriolis force. There may also be a modified pressure field $p(x, y)$. If the slowed wind reaches a new state of geostrophic balance, the cross wind components of Eq. (1) would reduce to:


$$\frac{\nabla p}{\rho} = -f \, \vec{k} \times \vec{u} \, . \tag{4}$$

## 3.3 Momentum mixing and Rayleigh Friction

The vertical mixing process is difficult to model. The eddies causing the vertical transport of momentum may be ambient or
"wake-generated" and may be sensitive to buoyancy effects in the boundary layer. Vertical mixing may be suppressed in a stable boundary layer or enhanced in an unstable one. The boundary layer stability in turn is influenced by surface heat flux and any warming aloft caused by broad tropospheric descent or cumulus convection (Barstad 2016). In this paper, we represent the complex vertical mixing processes by a simple Rayleigh friction. Rayleigh friction decays the wind disturbance at a rate proportional to its local value. The *ad hoc* nature of the Rayleigh friction approach makes it difficult to estimate values of the
coefficient $C$. Using a skin friction method, Smith (2010) chose $C=0.0001$ s$^{-1}$, noting that here we combine the upper and lower Rayleigh coefficients as $C = C_T + C_B$. Gribben and Adams (2023) used estimates for $C_B$ and $C_T$ in a manner that is sensitive to surface layer stability via the surface layer friction velocity $u_*$ as follows. $C_B$ can be estimated as (Smith 2007):

$$C_B = 2(u_*)^2/(HU) \tag{5}$$






where $H$ is the atmospheric boundary layer (ABL) height. By then assuming that the upper and lower friction forces are approximately equal in magnitude, $C_T$ can be estimated as (Smith 2007):

$$C_T = C_B U / (U_g - U) \tag{6}$$


where $U_g$ is the geostrophic wind speed. Alternatively, the observed length of wakes could be used to "reverse-engineer" a value for $C$.

### 3.4 Wake Recovery Integrals

We can learn about the Coriolis and Rayleigh contributions to wake recovery from the governing equation (1) by integrating

over the whole domain (see Smith 2022). The x-momentum Eq. (1a) gives, for westerly flow ($V$=0)

$$0 = \iint F_X \, dx dy \; - C \iint u(x,y) dx dy \; + f \iint v(x,y) dx dy \tag{7}$$

Note that the other terms in Eq. (1a) integrate to zero if the disturbance velocity and pressure decay at infinity. We define the

global Fractional Coriolis Recovery (FCR) and Fractional Rayleigh Recovery (FRR) as the fraction of the wake recovery due to Coriolis or Rayleigh forces respectively.

$$FCR = \; f \iint v(x,y) dx dy / \iint F_X \, dx dy \tag{8a}$$

and

$$FRR = -C \iint u(x,y) dx dy \; / \iint F_X \, dx dy \tag{8b}$$

From Eq. (7) we have

$$FCR + FRC = 1 \tag{9}$$


so together, the Coriolis and Rayleigh forces balance the net upstream turbine drag from the farm.

Another useful diagnostic is the Coriolis contribution to wake recovery along a streamline. For this purpose, we temporarily neglect the action of PGF, Rayleigh friction and diffusivity. Integrating Eq. (1a) downstream of the wind farm for westerly

flow ($V = 0$) gives the net Coriolis Recovery (CR) in units of ms[-1]





$$CR(x,y) = f\Delta(x,y) \tag{10}$$

where $\Delta(x,y)$ is the lateral displacement of a fluid parcel, given by


$$\Delta(x,y) = U^{-1} \int_{-\infty}^{x} v(x,y)dx \tag{11}$$

Physically, every increment of lateral displacement creates a downstream Coriolis acceleration, helping to restore the wake. Thus, $\Delta$ is a measure of the Coriolis Recovery. For example, if turbine drag slows the wind by 1 ms⁻¹, it can be recovered by

Coriolis force alone (i.e. $CR = 1$ ms⁻¹) with a $\Delta = \frac{CR}{f} = 10$ km lateral displacement in the case $f = 0.0001$ s⁻¹.

## 4 Idealized 1-D solution with no pressure field

### 4.1 Damped Inertial Waves

A simple one-dimensional solution to Eq. (1) might arise from a westerly flow across a thin row of turbines with $F_y = V = K = 0$ with $p(x) = 0$. Then using delta function forcing


$$F_x(x) = B\delta(x) \tag{12}$$

gives $u(x) = v(x) = 0$ upwind and


$$u(x) = \left(\frac{B}{U}\right) exp\left(-\frac{Cx}{U}\right) \cos\left(\frac{fx}{U}\right) \tag{13a}$$

$$v(x) = -\left(\frac{B}{U}\right) exp\left(-\frac{Cx}{U}\right) \sin\left(\frac{fx}{U}\right) \tag{13b}$$

downwind. The factor $B$ (with units m²s⁻²) is the integrated turbine drag across the farm. Solution Eq. (13) is a standing inertial wave with a restoring Coriolis force and damping by Rayleigh friction. In the case $f = 0$, $v(x) = 0$ the speed deficit Eq. (13a)

decays according to the Raleigh decay length $L_{RAY} = U/C$. For example, with $U = 10$ ms⁻¹ and $C = 0.0001$ s⁻¹, $L_{RAY} = 100$ km. With $f = 0.0001$ s⁻¹, wake recovery is somewhat faster due to the Coriolis force contribution, reaching $\frac{u(x)}{u(0)} = e^{-1}$ at $x = 72$ km, a 28% shortening of the wake. This formulation is useful in understanding the infinitely wide wind farm cases investigated numerically by Maas (2023).



## 4.2 Global Recovery

The global Fractional Coriolis and Rayleigh Recoveries are found by substituting Eq. (13) into Eq. (8a,b) giving

$$FCR = \frac{1}{1+\left(\frac{C}{f}\right)^2} \quad \text{and} \quad FRR = \frac{1}{1+\left(\frac{f}{C}\right)^2} \tag{14a,b}$$

satisfying Eq. (9). For example, if $C = f$, then $FCR = 1/2$ and half the wake recovery is caused by the Coriolis force. A more general derivation of Eq. (14) will be seen in Section 5.

### 4.3 Lateral deflection

Another way to diagnose the Coriolis Recovery is to compute the lateral parcel displacement by putting Eq. (13b) into Eq. (11) giving

$$\Delta(x) = -\left(\frac{B}{U}\right)\left[\frac{f-\left(C\sin\left(\frac{fx}{U}\right)+f\cos\left(\frac{fx}{U}\right)\right)\exp\left(-\frac{Cx}{U}\right)}{C^2+f^2}\right] \tag{15}$$

This lateral displacement oscillates but eventually decays to

$$\Delta(x \to \infty) = -\left(\frac{B}{U}\right)\frac{f}{C^2+f^2} \tag{16}$$

According to Eq. (16), increasing Rayleigh friction ($C$) reduces the final lateral displacement by damping the inertial wave before it completes its natural oscillation. Using Eq. (10), this gives an FCR in agreement with Eq. (14).

These special solutions, Eqns. (13-16), are helpful in understanding the competition between Coriolis and Rayleigh forces and the role of lateral streamline deflection, but they miss key aspects of wake dynamics. Missing are the roles of finite farm width, the disturbed pressure field and the tendency for the wake to approach geostrophic balance. To include these essential aspects, we solve Eq. (1), including the pressure field, using double Fourier transforms (Smith 2010).



# 5 Fourier Solution Methods

## 5.1 Turbine Layer

In Fourier space, the governing equations, Eqn. (1), become (with air density $\rho$ hidden in $p$)


$$ikU\hat{u} + ilV\hat{u} = \widehat{F}_x - ik\hat{p} - C\hat{u} - K(k^2 + l^2)\hat{u} + f\hat{v} \tag{17a}$$

$$ikU\hat{v} + ilV\hat{v} = \widehat{F}_y - il\hat{p} - C\hat{v} - K(k^2 + l^2)\hat{v} - f\hat{u} \tag{17b}$$

where $k$ and $l$ are the wavenumber vector components. These equations are shortened by defining the complex acceleration-
friction-diffusion operator

$$D(k,l) = ikU + ilV + C + K(k^2 + l^2) \tag{18}$$

so Eq. (17) becomes

$$D\hat{u} = \widehat{F}_x - ik\hat{p} + f\hat{v} \quad \text{and} \quad D\hat{v} = \widehat{F}_y - il\hat{p} - f\hat{u} \tag{19a,b}$$


We solve these two simultaneous equations for $\hat{u}(k,l)$ and $\hat{v}(k,l)$ by substituting and grouping terms to obtain

$$\hat{u}(k,l) = [\widehat{F}_x - ik\hat{p} + \frac{f}{D}(\widehat{F}_y - il\hat{p})]/\left[D + \frac{f^2}{D}\right] \tag{20a}$$

$$\hat{v}(k,l) = [\widehat{F}_y - il\hat{p} - \frac{f}{D}(\widehat{F}_x - ik\hat{p})]/\left[D + \frac{f^2}{D}\right] \tag{20b}$$


The inertial waves described by Eq. (13) can be seen in the Fourier space representation Eq. (19, 20). If there is no dissipation
(i.e. $C = K = 0$), the operator $D$ becomes $D = i\sigma = i(Uk + Vk)$ where $\sigma$ is the intrinsic frequency (i.e. the frequency seen by
an air parcel). The inertial waves occur when $\sigma = \pm f$. Note that the square bracket in the denominator of Eq. (20) vanishes in
this case. This singularity indicates a "free mode" where a widespread disturbance can exist with just local forcing.

## 5.2 Pressure forces and the upper layer

To complete the analysis, we include the hydrostatic pressure field generated by density anomalies aloft. The pressure
anomalies are created by vertical displacement $\eta(x, y)$ of the inversion layer according to

$$\hat{p}(k,l) = (g' + \frac{iN^2}{m})\hat{\eta} = \Phi\hat{\eta} \tag{21}$$




where $g'$ and $N^2$ are stability parameters for the inversion and free troposphere respectively (Smith, 2010). The quantity $m(k, l)$ in Eq. (21) is the vertical wavenumber for inertial gravity waves

$$m(k, l) = \frac{\pm N(k^2 + l^2)^{1/2}}{(\sigma^2 - f^2)^{1/2}} \tag{22}$$


Note that we have chosen to include the Coriolis parameter in this vertical wavenumber formulation. This is an essential feature as we are including the Coriolis term in the turbine layer formulation too. Noting the sign ambiguity in Eq. (22), we break the wavenumber spectrum into two parts. When $\sigma^2 > f^2$, we have inertial gravity waves and choose the sign from the radiation condition: $sgn(m) = sgn(\sigma)$. The phase lines tilt upwind with height. When $f^2 > \sigma^2$, $m$ is imaginary, the disturbance is
evanescent and we chose the decaying (i.e. positive imaginary) root. This two-part approach is well established in the literature, see for example see Smith 1979, 1982, Sutherland 2010, Nappo 2012.

To couple the disturbance in the lower and upper layers, we compute the vertical displacement $\eta(x, y)$ of the inversion at $z = H$. We do this with the continuity condition in the lower layer

$$w(z = H) = U \frac{\partial \eta}{\partial x} + V \frac{\partial \eta}{\partial y} = -H \left( \frac{\partial u}{\partial x} + \frac{\partial v}{\partial y} \right) \tag{23}$$

which in Fourier space is

$$\sigma \hat{\eta}(k, l) = -H(k\hat{u} + l\hat{v}) \tag{24}$$


It is important to note that the wave disturbance aloft and the disturbance in the lower turbine layer each influence the other. Thus, Eq. (24) must be solved simultaneously with Eqs. (20, 21). Doing this, the vertical displacement of the inversion becomes

$$\hat{\eta}(k, l) = \frac{-H[k(D\widehat{F_x} + f\widehat{F_y}) + l(D\widehat{F_y} - f\widehat{F_x})]}{\sigma(D^2 + f^2) - iDH(k^2 + l^2)\Phi} \tag{25}$$


When $f = 0$, Eq. (25) reduces to

$$\hat{\eta}(k, l) = \frac{-H(k\widehat{F_x} + l\widehat{F_y})}{\sigma D - iH(k^2 + l^2)\Phi} \tag{26}$$

which agrees with Eq. (4) in Smith (2010). The turbine layer velocity perturbations $\hat{u}(k, l)$ and $\hat{v}(k, l)$ are found by
substituting Eqs. (21, 22, 25) into Eq. (20) so





$$\hat{u}(k,l) = \frac{D\widehat{F_x} + f\widehat{F_y} - i\Phi(Dk+fl)\hat{\eta}}{(D^2+f^2)} \tag{27a}$$

$$\hat{v}(k,l) = \frac{D\widehat{F_y} - f\hat{F_x} - i\Phi(Dl-fk)\hat{\eta}}{(D^2+f^2)} \tag{27b}$$

where $D(k,l)$ is given by Eq. (18). Using the inverse Fast Fourier Transform (FFT), the fields $u(x,y)$, $v(x,y)$ and $\eta(x,y)$ are found. Eqs. (27, 28) capture a wide variety of fluid dynamical processes such as upstream blockage and deflection, vortex stretching, inertial waves, shallow water waves, vertically propagating gravity waves, frictional dissipation, lateral momentum diffusion and geostrophic adjustment. One disadvantage of the FFT solution is that the solutions are assumed to be periodic and thus can wrap from the exit to the entrance region if the Rayleigh friction or domain size are insufficient.


**5.3 Global Wake Recovery**

The Fourier solutions (see Eq. 25, 27) can be used to find the global Fractional Coriolis Recovery (FCR, see Eq. (8a)) using the property of Fourier transforms that the area integral of any function $G(x,y)$ is given by its Fourier Transform $\hat{G}(k,l)$

evaluated at $k = l = 0$; that is

$$\iint G(x,y)dxdy = \hat{G}(k=0, l=0) \tag{28}$$

except for a possible normalizing coefficient. Using Eq. (18)

$$D(k \to 0, l \to 0) = ikU + ilV + C + K(k^2 + l^2) \to C \tag{29}$$

so that Eqs. (27, 28) gives (assuming a westerly flow and no lateral forcing)

$$\hat{u}(k=0, l=0) = \frac{C\widehat{F_x}(k=0,l=0)}{(C^2+f^2)} \tag{30a}$$


$$\hat{v}(k=0, l=0) = \frac{-f\hat{F_x}(k=0,l=0)}{(C^2+f^2)} \tag{30b}$$

The global Fractional Coriolis Recovery is then

$$FCR = \frac{-fv(k=0,l=0)}{\hat{F}_X(k=0,l=0)} = \frac{f^2}{(C^2+f^2)} = \frac{1}{1+(\frac{C}{f})^2} \tag{31}$$

and the Fractional Rayleigh Recovery is



$$FRR = \frac{Cu(k=0,l=0)}{\hat{F}_X(k=0,l=0)} = \frac{C^2}{(C^2+f^2)} = \frac{1}{1+(\frac{f}{C})^2} \tag{32}$$

both in perfect agreement with Eq. (14). Thus, we learn that global FCR and FRR are not altered by finite warm width, stratification effects or lateral dispersion effects. However, the reader should be alert to the fact that these global measures of recovery do not provide information on wake recovery at a specific location so are of limited use on their own for practical wake studies.

### 5.4 Diagnostic Fields

The impact of the Coriolis force and Rayleigh friction on the wake recovery can be seen using three diagnostic fields. The scalar wind speed deficit field is (Smith 2022)

$$Deficit(x,y) = -\frac{\vec{U} \cdot \vec{u}}{|\vec{U}|} = -(Uu + Vv)/(U^2 + V^2)^{1/2} \tag{33a}$$

This definition of Deficit is simpler if we use natural coordinates $(x', y')$ aligned with and perpendicular to the ambient wind direction and the corresponding perturbation wind components $(u', v')$ so that

$$Deficit(x',y') = -u'(x',y') \tag{33b}$$

The scalar cross wind speed is

$$Crosswind(x,y) = \frac{(\vec{U} \times \vec{u}) \cdot \vec{k}}{|\vec{U}|} = (Uv - Vu)/(U^2 + V^2)^{1/2} \tag{34a}$$

or
$$Crosswind(x',y') = v'(x',y') \tag{34b}$$

where $\vec{k}$ is the unit vector in the vertical direction. Only two processes create a Crosswind, assuming that the turbine drag opposes the ambient wind. The high pressure region upwind of the farm will deflect air left and right giving a pair of Crosswind regions of opposite sign. If the turbine drag slows the air, the excess Coriolis force will deflect air leftward in the wake. The Crosswind is an important diagnostic of the Coriolis force impact, see Eqs. (10, 11).

The vertical displacement of the inversion $\eta(x,y)$ in Eq. (24) is also a useful diagnostic as it provides information on the divergence in the turbine layer and the forcing of gravity waves aloft that imprint a pressure field on the lower layer, see Eqs.



(21, 25). An interesting property of these three diagnostics is their left-right symmetry across the wake. Cross-wake symmetry is judged relative to the centerline; a line parallel to the ambient flow passing through the farm center. This symmetry can be

determined from the solutions Eqs. (25, 27) in Fourier space as even/odd functions have even/odd Fourier Transforms. The result of such a symmetry analysis is shown in Table 1 and can be seen in Figures 1-3. The Rigid Lid case is described in the Appendix.

**Table 1** Cross Wake Symmetry for a Symmetric Wind Farm


| Diagnostic | $f = 0$ | $f \neq 0$ | $f \neq 0$ Rigid Lid |
|---|---|---|---|
| Vertical Displacement | symmetric | non-symmetric | non-symmetric |
| Deficit | symmetric | symmetric | symmetric |
| Crosswind | anti-symmetric | non-symmetric | anti-symmetric |

## 6 Fast Fourier Transform (FFT) Wake Computations

Analysis of the wake structure requires specification of the turbine forces acting on the lower layer. Here we use

$$\vec{F}(x,y) = -A\frac{\vec{U}}{|\vec{U}|}exp(-\frac{x^p+y^p}{a^p})  \qquad (35)$$

where $A$ is the drag in acceleration units at the farm center. When exponent $p = 2$, Eq. (35) gives a smooth circular Gaussian

force field but we use $p = 20 >> 1$ so Eq. (35) gives a sharp-edged square wind farm with dimensions $2a$ by $2a$. A "reference" wind deficit profile is found for westerly wind ($V = 0$) by integrating Eq. (1a) and Eq. (35) using the turbine drag force alone (i.e. $p(x,y) = f = C = K = 0$ ). This procedure gives

$$Deficit_{REF}(y') \approx D_0 \ \ for \ |y'| \leq a$$
$$Deficit_{REF}(y') \approx 0 \ \ \ for \ |y'| > a  \qquad (36)$$


where $D_0 = \frac{2Aa}{|\vec{U}|}$ and $y'$ is the distance from the centerline. If a wake deficit of $D_0 = 1$ ms$^{-1}$ is desired with a wind speed of $U = 10$ ms$^{-1}$ and farm half-width $a = 20$ km, we choose $A = 0.00025$ ms$^{-2}$. The area integrated force from Eq. (35) is then


$$\iint |\vec{F}| \, dxdy = 4Aa^2.  \qquad (37)$$





As $F$ and $A$ are expressed in acceleration units, the total farm drag in Newtons is written $4\rho H a^2 A = 192 \times 10^6$ N using values from Table 2. Using Eqs. (25, 27, 35) we computed wind farm diagnostic fields for three cases (see Figs. 1-5) using the parameters in Tables 2 and 3. Our domain has 800 by 800 grid points with a grid spacing of 1000 m. The calculation is quick
due to the efficiency of the FFT algorithm.

**Table 2** Model Parameters

| Parameter | Symbol | Units | Value |
|---|---|---|---|
| Wind speed | $|U|$ | ms$^{-1}$ | 10 |
| Farm Half Width | $a$ | km | 20 |
| Farm Drag | $A$ | ms$^{-2}$ | 0.00025 |
| Turbine Layer Depth | $H$ | m | 400 |
| Inversion strength | $g'$ | ms$^{-2}$ | 0.1 or 10.0 |
| Troposphere Stability | $N$ | s$^{-1}$ | 0.01 |
| Lateral Diffusivity | $K$ | m$^2$s$^{-1}$ | 200 |
| Rayleigh coefficient | $C$ | s$^{-1}$ | 0.0001 |
| Coriolis parameter | $f$ | s$^{-1}$ | 0 or 0.0001 |


**Table 3** Three Cases

| Case # | Coriolis Parameter, $f$ | Reduced Gravity, $g'$ | Conditions | $f/C$ | $FS = a/RRD$ | Figures |
|---|---|---|---|---|---|---|
| a | 0 | 0.1 | No Coriolis, typical stability | 0 | 0 | 1a-5a |
| b | 0.0001 | 0.1 | Coriolis, typical stability | 1 | 0.32 | 1b-5b |
| c | 0.0001 | 10 | Coriolis, strong stability | 1 | 3.2 | 1c-5c |

In Fig. 1 we show the wake Deficit (see Eq. (33)) for the three cases in Table 3. All three patterns are symmetric across the centerline (see Table 1). Cases b and c have small regions of negative deficit (i.e. wind speed above ambient) to the left and
right of the wake. The reason for these "edge jets" is discussed in Section 8. Figure 2 shows the corresponding Crosswind patterns (see Eq. (34)). In Case a we see the left and right upstream cross flow caused by the gravity wave pressure field. In Case b, the pattern is asymmetric as the Coriolis force deflects air leftward. In Case c, with strong stratification, the Coriolis deflection on the centerline is suppressed by the quick establishment of geostrophic balance (Section 8).





**Figure 1**

Wind speed deficit contour plots for three cases (see Table 3). Fields come from full FFT calculations of flow in the lower turbine layer. The wind direction is east to west (left to right). Dashed square marks the wind farm. See Table 1 for symmetries and Table 2 for common parameter values. These are zoomed in views, with the full domain extent being far greater than shown.


Figure 3 shows the vertical displacement of the inversion (see Eq. (25)). In Case a, we see the upwind lifting of the inversion that causes high pressure there. In Case b, note the important asymmetry across the wake, with lifting on the left and sinking on the right. This lateral gradient creates a cross wake PGF (Section 8). In Case c, the cross wake inversion tilt is still present.

The magnitude of vertical displacement is small now but the cross wake PGF is still strong. This PGF keeps the flow in geostrophic balance.







**Figure 2**

Similar to Figure 1 but for Crosswind, for Cases a,b,c in Table 3.






**Figure 3**

Similar to Figure 1 but for inversion layer vertical displacement, for Cases a,b,c in Table 3. Part (c) has an amplified scale.

## 7 Forces on the centreline

To understand the wake recovery more fully, we consider the forces acting on the air along the flow centerline. We substitute the computed $u(x, y)$ and $v(x, y)$ along the centerline into the RHS of Eq. (1) to find the perturbation forces there. We neglect the small effect of lateral momentum diffusion. In Figure 4, we show the streamwise forces acting on the air along the centerline. For Case a with $f = 0$ (Fig. 4a) air approaching the farm first feels a retarding pressure gradient force (PGF). Soon after, the large turbine drag force begins to act (see Eq. (35)). Near the farm center, the PGF quickly turns positive and helps to keep the air moving against the strong turbine drag. By this position, the wind speed deficit has become large and the Rayleigh friction is working hard to restore the wind speed. Rayleigh friction remains active far downwind. Case b with





Coriolis force acting (Fig. 4b) is similar but the Coriolis provides a significant positive force helping the wake to recover. In Case c, $f$ is still non-zero but there is no Coriolis recovery as there is no cross wind (Fig 2c).



**Figure 4**

Downstream force components at the centerline, for three cases (see Table 3).


The cross-wake forces acting on the centerline are shown in Fig 5. In the Case a with $f=0$, there are no cross-wake forces. In Case b (Fig. 5b) the slowed wake flow creates a leftward perturbation Coriolis force. Further downstream (say $x = 150$ km), the lateral PGF puts the flow back into geostrophic balance (see Eq. (3)). For Case c, (Fig. 5c) geostrophic balance develops immediately.






**Figure 5**

Cross stream force components at the centerline, for three cases (see Table 3).





## 8 Geostrophic Balance in the Wake

**8.1 Geostrophic Adjustment**

One of the most striking aspects of the FFT solutions is the quick adjustment to cross-wake geostrophic balance (See Eq. (4)) in the wake (Fig 5b, c). We estimate the adjustment distance $X_G$ as follows, using order of magnitude arguments. From Eq. (1b), slowed wake air will develop a leftward velocity $v(x) \sim -fux/U$ and from Eq. (11), a growing leftward displacement $\Delta(x) \sim -fux^2/U^2$. Here, the perturbation wind x-component $u = -Deficit$ from Eq. (33) and $x$ is the distance downwind of the farm center, and we neglect the Rayleigh friction. This leftward deflection distorts the inversion height $\eta(y) \sim H\left(\frac{\Delta}{a}\right)$ (from Eq. (23)) and produces a lateral pressure gradient $\frac{dp}{dy} \sim g'\frac{d\eta}{dy} \sim g'\left(\frac{\eta}{a}\right)$ (from Eq. (21), keeping only the inversion stability $g'$). The PGF continues to grow until geostrophic balance is reached $fu = -\frac{dp}{dy} \sim \frac{g'Hfux^2}{U^2a^2}$ or

$$\frac{X_G}{a} \approx Fr \qquad (38)$$

where the Froude number $Fr = U/\sqrt{g'H}$ and $a$ is the farm width. The Froude number also characterizes the shallow water waves in the solution and whether the flow is sub- or super-critical (Smith 2010). Surprisingly, this distance $X_G$ depends only on the wind speed, static stability and farm width. The Coriolis parameter $f$ cancels out of the estimate because the rate of deflection and the deflection needed for balance are both proportional to $f$. The strength of the wake deficit also cancels out. If the inversion stability $g' = 0$, the tropospheric stability $N$ plays a similar role (see Eq. (21)), but is more difficult to quantify (Smith 2024). Under typical atmospheric stability conditions (Table 4), $X_G$ is only about two farm widths downstream even if $f$ is very small. According to Eq. (38), with an infinitely wide farm, geostrophic balance could never occur (see Section 4) .

**8.2 Geostrophic Balance**

Once established, geostrophic balance requires that the perturbation cross-wake PGF and Coriolis forces cancel, see Eq. (4). Again neglecting the tropospheric stability $N$, we can write the cross-wake PGF as the product of reduced gravity $g'$ and the lateral inversion tilt.

$$-g'\frac{d\eta(y)}{dy} + f \cdot Deficit(y) = 0 \qquad (39)$$

Assuming that the inversion displacement $\eta(y)$ vanishes at infinity, integrating Eq. (39) requires that the net wake deficit vanish once geostrophic balance is established, i.e.



$$\int_{-\infty}^{\infty} Deficit(y) \, dy = 0 \tag{40}$$

Combining Eq. (39) with the continuity equation, Eq. (23)

$$\eta(y) = -H \frac{d\Delta}{dy} \tag{41}$$

and the Coriolis recovery formula Eqs. (10, 11)

$$Deficit(y) = Deficit_{REF}(y) - f\Delta(y) \tag{42}$$

gives a second order differential equation


$$g'H \frac{d^2\Delta}{dy^2} - f^2\Delta(y) = -f \cdot Deficit_{REF}(y) \tag{43}$$

for the lateral streamline deflection $\Delta(y)$ profile. The quantity $Deficit_{REF}$ is the initial wake deficit profile just behind the farm caused by turbine drag. A "box-car" wake Eq. (35) of width $2a$ has a constant $Deficit_{REF}$ inside the wake ( $|y| < a$ ) and

zero deficit outside the wake.  Requiring smoothness at the wake edges ($y = \pm a$) and decay at infinity, the solutions to Eq. (43) in the outer and inner wake are

$$\Delta(y) = A_1 \, exp(-\alpha|y|) \qquad\qquad \text{for } |y| > a \tag{44a}$$
$$\Delta(y) = A_2\big(exp(\alpha y) + exp(-\alpha y)\big) + B \qquad \text{for } |y| < a \tag{44b}$$

where the coefficients are

$$A_1 = \left(\frac{1}{2}\right)\left(exp(\overline{FS}) - exp(-\overline{FS})\right)B$$

$$A_2 = -\left(\frac{1}{2}\right) exp(-\overline{FS}) \, B$$

$$B = \left(\frac{1}{f}\right) Def_0$$


and where $\alpha = \left(\frac{f}{\sqrt{g'H}}\right) = RRD^{-1}$ and the non-dimensional farm size is $\overline{FS} = a/RRD$.  The Rossby Radius of Deformation (RRD) is a "communication distance" related to stratification and rotation.  On the centerline ($y = 0$), Eq. (44) with Eqs. (8a, 10) gives Fractional Coriolis Recovery





$$FCR = 1 - \exp(-\overline{FS}) \tag{45}$$

valid for $N=C=0$. Other wake variables can be computed from Eq. (44). The speed deficit profile comes from Eq. (42) and the vertical displacement of the inversion from Eq. (41). On the left side of the wake (looking downwind) the inversion is lifted while the right side it is depressed.


The sensitivity of the lateral deflection profile $\Delta(y)$ to $\overline{FS}$ is shown in Figure 6. With large $\overline{FS}$, the lateral deflection (and thus the Coriolis Recovery) acts primarily within the inner wake. Note however that the $\Delta(y)$ extends into the outer wake where the air was not slowed by the farm. The result is a narrow strip of air moving faster than ambient. We call this strip the "edge jet". Its magnitude is $f\Delta(y = a)$ from Eq. (44). When $\overline{FS} < 1$, the lateral deflection is small and widespread. The "outer

wake" air is pushed/pulled leftward by the "inner wake" air. The impact of geostrophic adjustment in this case is not to recover the inner wake but to accelerate the outer wake slightly above ambient. The net wake deficit is zero, see Eq. (40).

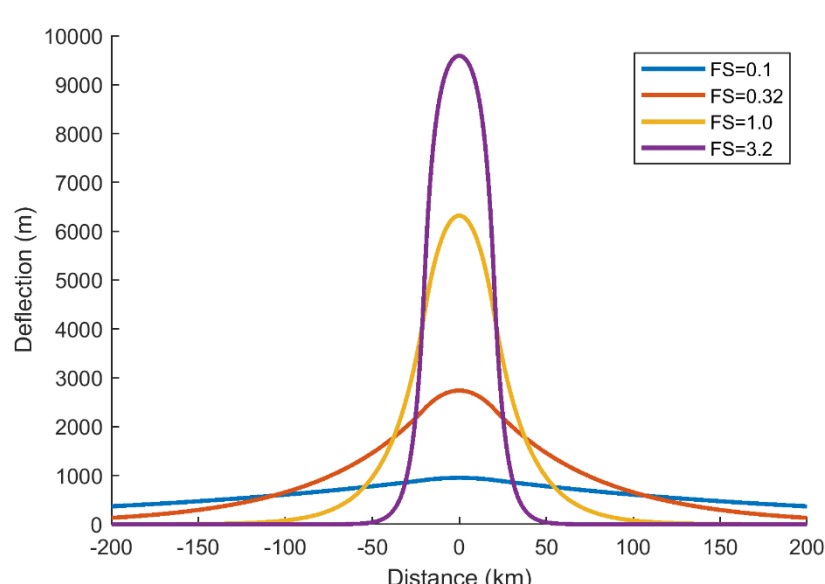

**Figure 6**

Sensitivity of cross-wake deflection, to achieve geostrophic balance, to non-dimensional farm size. The reference wake is 40km wide. For smaller $\overline{FS}$ , the deflection is smaller but more widespread. The area under each curve is the same.


### 8.3 Comparison of geostrophic theory with the full FFT model

While we argue that geostrophic adjustment plays an important role in wake recovery, the real world and the FFT model includes other processes such as pressure gradients from vertically propagating gravity waves, shallow water waves and

Rayleigh friction. Here we compare the Deficit and Vertical Displacement profiles from geostrophic theory Eq. (44) with a




full FFT model run at $x$=75 km downwind of the farm center (Fig 7). We use our "standard" model run with $f = C = 0.0001$ s$^{-1}$, $g' = 0.1$ ms$^{-2}$, $N = 0.01$ s$^{-1}$ and $\overline{FS} = 0.316$ (Tables 2,3). The position $x = 75$ km is chosen from Figure 5 as a point with geostrophic balance and still a strong wake deficit. The agreement in Figs 7 is good and improves if we increase $f/C$.


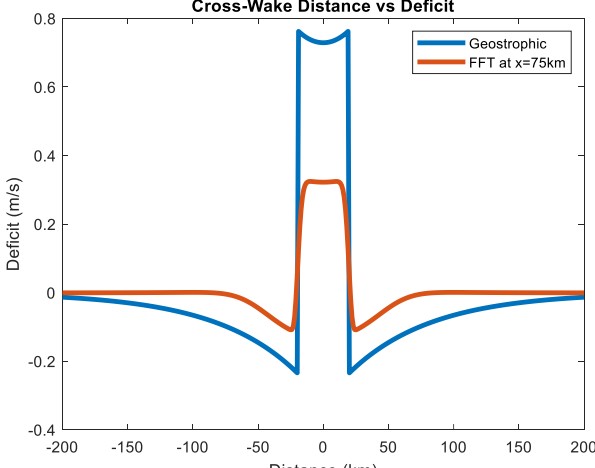
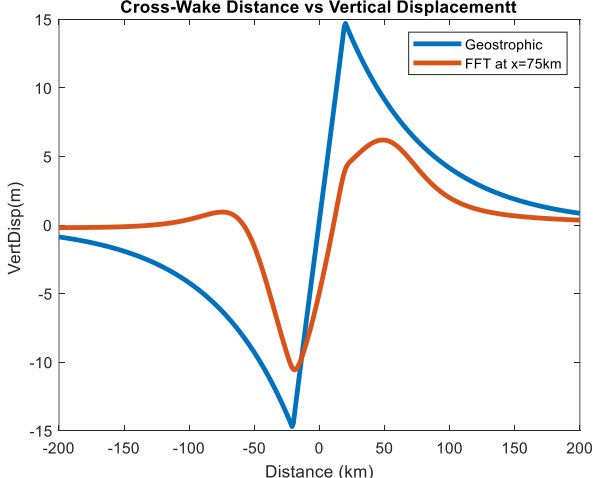

**Figure 7.** Comparison of geostrophic theory (Eq. 44) and FFT solutions, (a) for wake deficit and (b) for vertical displacement of the inversion layer. Deficit profiles show a local minimum on the centreline (Distance $y = 0$) and edge jets near $y = \pm a$. Vertical displacement profiles show extrema of about 10 meters near the wake edges and a strong tilt across the inner wake. This tilt causes the PGF that balances the remaining wake deficit.

To further compare geostrophic theory with the FFT model, we chose the maximum FCR on the centerline as a measure of Coriolis recovery , see Eq.(45). This quantity is plotted in Figure 8 against the two non-dimensional control parameters $f/C$ and $\overline{FS}$. For small $f/C$, Raleigh friction generally dominates as it recovers the wake before Coriolis can act. For larger $f/C$, maximum FCR is sensitive to farm size $\overline{FS}$. With small $\overline{FS}$, the stratification quickly establishes geostrophic balance and FCR is small. With large $\overline{FS}$, the Fractional Coriolis Recovery (FCR) is more significant. This sensitivity to $\overline{FS}$ is captured in Eq (45). The Global FCR is always greater than centreline FCR.



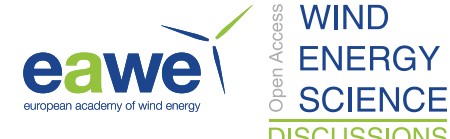

**Figure 8**

Sensitivity of maximum Fractional Coriolis Recovery on the wake centreline to the ratio $f/C$ and to farm size $\overline{FS}$. A global threshold (i.e. not only in the wake) is provided by Eq. (31), and Eq. (45) provides an estimate of wake centreline FCR in the absence of Rayleigh friction.

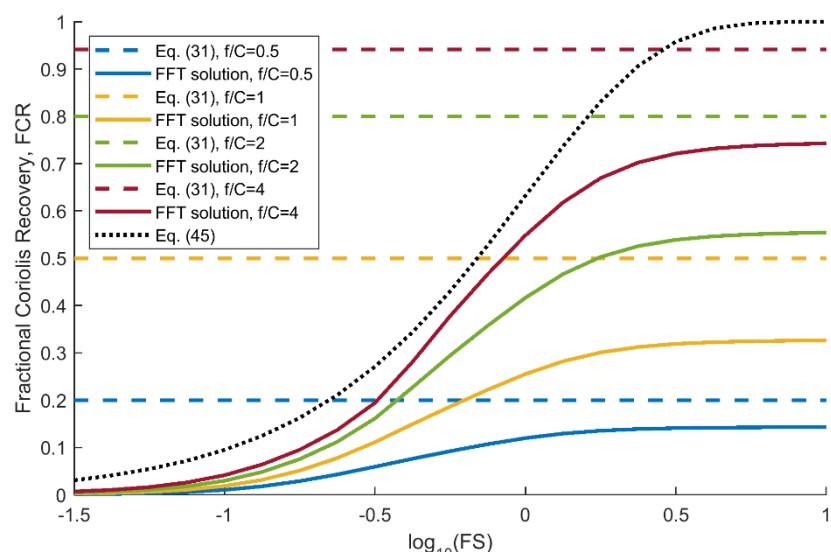

## 9. Applications

In this section we consider how the current analyses in this paper apply to the real world.

### 9.1 Non-dimensional farm size and Froude Number

Three non-dimensional parameters control most of the results in this paper: $C/f$, $Fr$ and $\overline{FS}$. Of these, $C/f$ is the most difficult to estimate due to the uncertainty in the Rayleigh coefficient $C$. In Table 4 we use a selection of atmospheric characteristics and wind farm sizes to estimate the range of the other two parameters: $Fr$ and $\overline{FS}$. We fix $f = 0.0001$ s$^{-1}$ corresponding to a latitude of about 45 degrees. We neglect the contribution from the continuous stratification above the boundary layer ($N$), which serves to strengthen the effect of the inversion. The small and large wind farm areas used are for Horns Rev 1, and for Hornsea 1 and Hornsea 2 combined, respectively. The wind farm radius is derived by simply considering these areas as circles with radius $a$. The values selected for $\Delta\theta$ are supported by radiosonde data analysis from Barstad 2015 which was broadly replicated in a study by Gribben et al. 2023 although not included in that publication. The large and small values for $H$ are derived in the same way, although the typical $H$ value comes from Rodaway et al. 2024.





Table 4 Ranges of non-dimensional length scale $\overline{FS}$. Note that Froude number values assume $U = 10$ ms[-1]

| Parameter | Symbol | Units | Value for Small $\overline{FS}$ | Value for Large $\overline{FS}$ | Value for Typical $\overline{FS}$ |
|---|---|---|---|---|---|
| Wind Farm Area | | km$^2$ | 19 | 869 | 100 |
| Wind Farm Radius | $a$ | km | 2.46 | 16.63 | 5.64 |
| Inversion strength | $\Delta\theta$ | K | 5 | 0.5 | 1.5 |
| Reduced gravity | $g'$ | ms$^{-2}$ | 0.1721 | 0.01721 | 0.05163 |
| Layer depth | $H$ | m | 2000 | 300 | 500 |
| Coriolis parameter | $f$ | s$^{-1}$ | 0.0001 | 0.0001 | 0.0001 |
| Rossby Radius of Deformation | $RRD$ | m | 185.5 | 22.7 | 50.8 |
| Non-dimensional farm size | $\overline{FS}$ | none | 0.01 | 0.73 | 0.11 |
| Froude number | $Fr$ | none | 0.54 | 4.40 | 1.97 |

Even for the Large $\overline{FS}$ case, we can see from Table 4 that $\overline{FS} < 1$ therefore the wake recovery by Coriolis force will always be reduced by geostrophic balance, see Eq. (45). In the small $\overline{FS} = 0.01$ case, the rigid lid limit (Appendix 1) applies and geostrophic balance will prevent the Coriolis force from contributing to wake recovery.

The Froude number ranges from $Fr = 0.54$ to $Fr = 4.40$ in Table 4. These values imply that geostrophic balance will be
commonly achieved quickly behind wind farms, see Eq. (38).

The methods of this paper might also be applied to natural wakes caused by mountains, islands or irregular coast lines. Wakes from mountainous islands such as St Vincent in the Caribbean (Smith *et al.* 1997) and Hawaii (Smith and Grubisic, 1993) sometimes extend to 200 km. These natural wakes may be important for offshore wind farm siting.


### 9.2 When will Coriolis force be important?

As the magnitude of the Coriolis force on earth is generally small, it is fair to ask whether it can be important for wake recovery. Furthermore, if the Rayleigh force (i.e. momentum mixing) is large, it will dominate recovery before Coriolis can act (Section 4). We also know from Section 8 that in a stable atmosphere Coriolis recovery is often reduced by geostrophic adjustment,
especially for small farms. After some exploration of parameter space, we suggest that the most likely scenario for Coriolis impact (on inner wake recovery) is low wind, wide farm and weak stability in addition to small $C/f$.

A baseline test case to see Coriolis impact is constructed based on a square wind farm (half-width $a = 20$ km) with a uniform momentum sink, with a strength which on its own would result in a deficit of $Def_{REF} = 1$ ms[-1] (see Eq. (35)). Atmospheric
stability values are $g' = 0$ and $N = 0.01$ s[-1]. A second case was constructed with a four times greater wind farm half-width (80 km), having the same length and reference deficit. Each case was run with and without Coriolis forcing, with other conditions selected to emphasise the Coriolis effect while remaining realistic: $C = 0.00005$ s[-1], $f = 0.000124$ s[-1], freestream




wind speed = 7 ms⁻¹. The deficit profiles for each of the resulting runs, at 50 km downwind from the wind farm, are shown in Figure 9.


The impact of the Coriolis force on the inner wake deficit is evident by comparing the solid lines ($f = 0.000124$ s⁻¹) with the dotted lines ($f = 0$) in Figure 9, especially for the wider case.  Figure 9 also shows acceleration of the outer wake and symmetric edge jets. This large impact provides motivation for including the Coriolis force in operational wake models.


**Figure 9**

Deficit profiles at a distance 50 km downwind from the wind farm edge. FFT calculations are shown with ($f = 0.000124$ s⁻¹) and without Coriolis force modelling ($f = 0$). In each case the wind farm length is 40 km, with results for wind farm widths of 40 km and 160 km shown.

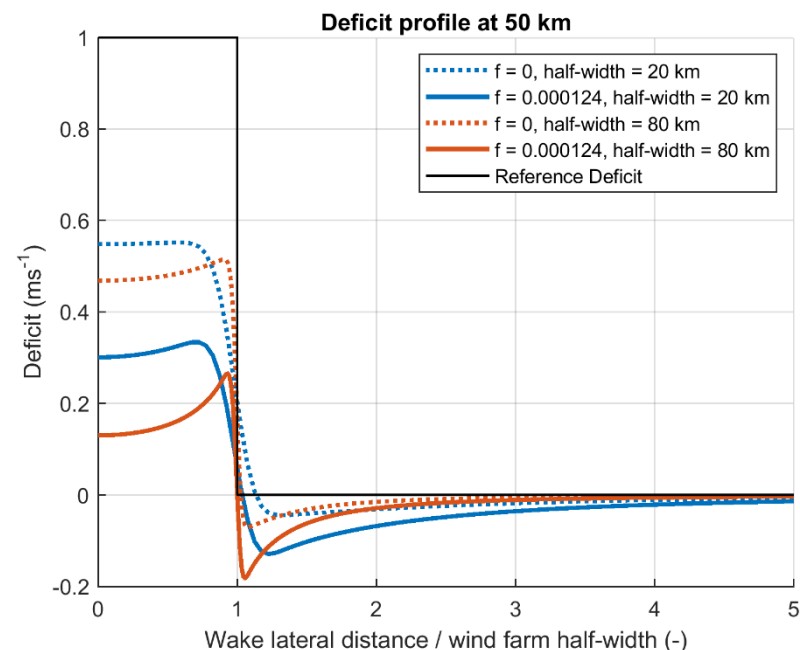

**10 Conclusions**

We examined wake recovery from the Coriolis and Rayleigh forces using a steady linearized two-layer model.  This model

allows us to obtain analytical expressions and do numerical wake computations including several interacting fluid dynamical processes.

In this complex problem, the simplest behaviour is the exponential recovery of the wake speed deficit by momentum mixing, parametrized as Rayleigh friction. This type of wake recovery gives an e-folding length scale of $L_{RAY} = U/C$ where $U$ is the

ambient wind speed and $C$ is the Rayleigh coefficient. For example, if $U = 10$ ms⁻¹ and $C = 0.0001$ s⁻¹, the e-folding length for

the wake is $L_c \approx 100$ km. When Coriolis force is added, it accelerates the wake recovery and shortens the wake by introducing a damped inertial wave.

An interesting measure of the Coriolis force impact is the global Fractional Coriolis Recovery (FCR) and its complement the
Fractional Rayleigh Recovery (FRR). When the ratio $C/f$ is decreased, the Coriolis force does more of the wake recovery and Rayleigh friction does less. The expressions for global FCR and FRR derived in the idealized 1-D model continue to hold in the complex 3-D model including stratification and pressure disturbances. However, these global measures do not tell us all that we need about local wake structures.

The key finding in the paper is the strong tendency for the wake to approach geostrophic balance. This balance occurs through a mutual adjustment of the wake deficit and the cross-wake pressure gradient. When the Coriolis force deflects wake air leftward (in the northern hemisphere) two changes to the wake occur: a reacceleration of wake air and a distortion of the pressure field. Together, these changes bring the wake air into balance. The nature of the balanced wake depends on the non-dimensional farm size $\overline{FS} = a/RRD$ where $a$ is the half-width of the farm and the Rossby Radius of Deformation (RRD) is a
measure of atmospheric stability. A geostrophic wake theory (Section 8.2) explains this dependence well. When $\overline{FS} > 1$, the Coriolis Recovery (CR) is effective at accelerating air in the "inner" wake. By pushing/pulling the adjacent "outer" wake leftward, it also creates narrow "edge jets" to the left and right of the wake. In the opposite case of $\overline{FS} < 1$, the CR in the inner wake is weak as most of the geostrophic adjustment occurs via the PGF rather than flow acceleration. In this case, the wake speed can only be recovered by Rayleigh friction. At the same time however, a weak widespread Coriolis acceleration occurs
over the outer wake. The model suggests that this far-reaching Coriolis acceleration might benefit off-axis downwind farms.

If our results are correct, future wake models should include the Coriolis force and outer wake acceleration, especially in cases with large farms, small wake mixing, weak atmospheric stability and high latitude.

**Code Availability**: The MATLAB code used in this paper is available from the first author.

**Data Availability:** There are no experimental data in this paper. The data for plots and tables come from MATLAB code.

**Competing Interests:** The authors have declared that there are no competing interests.


**Author Contributions:** BG proposed the project and asked RB for help in properly applying Coriolis in the upper layer's vertical wave number. The revised solution for the wind farm layer was a joint effort. RB led on all subsequent theoretical analyses, with BG confirming derivations and computations. Both authors developed software implementations independently as a cross-check.





**Acknowledgements**: We thank Nicolai Gayle Nygaard for insightful comments. We regret the recent passing of Dries Allaerts from the Delft University of Technology who worked to promote a deeper understanding of this subject and offered generous support to the authors' work with memorably knowledgeable and constructive review and discussions.

**Appendix 1 : The limit of strong stratification**

As argued in Section 8, strong atmospheric stratification weakens or eliminates Coriolis wake recovery leaving only Rayleigh frictional recovery. This conclusion can be demonstrated by considering the limit of $g' \to \infty$ making the inversion act like a rigid lid and making the turbine layer flow non-divergent (Smith 2024). This non-divergent limit is not really that extreme and in fact is well satisfied by the two real cases in Table 4 with small $\overline{FS} = 0.01$ and $0.11$.

To investigate the strong stratification limit, we take $g' \to \infty$ in Eqs. (25,27) giving new expressions for the velocity field in Fourier space

$$\hat{u}(k,l) = \frac{l^2 \hat{F}_X - kl\,\widehat{F_Y}}{D(k^2 + l^2)} \tag{A1a}$$

$$\hat{v}(k,l) = \frac{-kl\,\hat{F}_X + k^2 \hat{F}_Y}{D(k^2 + l^2)} \tag{A1b}$$

These simple expressions satisfy the non-divergent condition in Fourier space

$$\widehat{Div} = ik\hat{u} + il\hat{v} = 0 \tag{A2}$$

The Coriolis parameter $f$ cancels out in this derivation and does not appear in Eq. (A1), demonstrating the lack of Coriolis force influence on the perturbation velocity field. However, taking the same $g' \to \infty$ limit, the pressure field Eqs. (21,25) becomes.

$$\hat{p}(k,l) = \frac{(k\widehat{F_x} + l\hat{F}_Y)}{i(k^2 + l^2)} + \frac{f(k\,\hat{F}_Y - l\hat{F}_X)}{iD(k^2 + l^2)} \tag{A3}$$

in which $f$ still appears. The first term in Eq. (A3) is the dipole-like pressure field that decelerates and splits the airstream near the farm (Smith 2024). It is symmetric across the centreline and anti-symmetric along the flow direction with high/low pressure on the windward/leeward side of the farm. It is a local pressure response to the turbine drag. Note that this term includes no



flow parameters (i.e. no $U, V, C, K, f, g', H$). The second term in Eq. (A3) describes the pressure field in the wake. It is

proportional to $f$ and is anti-symmetric across the centerline. It geostrophically balances the wake deficit until Rayleigh

friction restores the wake. For example, with $F_Y = V = 0$, the north-south Coriolis force from (A1a)    $-f\hat{u}(k,l) =$

$-fl^2 \hat{F}_X / (D(k^2 + l^2))$ is equal and opposite to the north-south pressure gradient force from the wake term in Eq. (A3)

$-il\hat{p}(k,l) = fl^2 \hat{F}_X / \left(D(k^2 + l^2)\right)$. As these two expressions in Fourier space are equal and opposite, Eq. (4) is satisfied

everywhere in the wake, except for vestiges of the local turbine drag term. Note that we have taken the rigid lid limit with

$g' \to \infty$ but we probably could have done it with tropospheric stability $N^2 \to \infty$ too (Smith (2024)).

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
