# Peer review of "Coriolis Recovery of Wind Farm Wakes"

_Wind Energy Science, 2025_

## Author Comment (AC1)

These author comments are in response to reviewer comments "RC1". The original comments are quoted here in plain typeface for ease of reference, with author comments in bold.

This manuscript examines how the Coriolis force affects wind farm wake recovery using a linearized, steady-state, two-layer model with turbine drag applied to the lower layer. It extends previous work by the same author(s), which mainly focused on wind-farm-induced gravity waves and their effect on the surrounding pressure field. The strength of this approach is its computational efficiency, allowing quick analysis of large-scale flow features.

**Agreeing with this comment, we would also note that another strength of this approach is that it affords a theoretical analysis which leads to some useful insights**.

**The FFT approach gives us a useful tool, and the present work adds to the literature on this modelling approach by introducing the Coriolis term into the vertical wave number which is key to the upper layer pressure / displacement response. However the FFT approach is not the point of this paper. Our focus is the fluid dynamics of how the Coriolis force impacts the wake and how it interacts with pressure gradient forces. The reviewer does not mention any of our derived closed form expressions (e.g. 31,32,45), symmetry arguments (table 1) or new insights such geostrophic adjustments and the "edge jets". These expressions, arguments and insights are new to the literature on farm wakes.**

However, this modeling simplicity sacrifices some interpretability and accuracy. For instance, in Figure 7, the difference between the FFT model and geostrophic theory is roughly a factor of 2, suggesting limited accuracy.

**The authors agree that the difference between the FFT solutions and the 'geostrophic theory' in Figure 7 may be characterised as a factor of 2. The value in comparing the FFT solution to the geostrophic theory is in demonstrating that for all of the assumptions and simplifications that go into developing the 'geostrophic theory' in this context, the theory appears to be fairly successful in predicting the geostrophic adjustment-related response of the wake (as embodied by the full FFT solution). The qualitative response is well represented, and the quantitative match to within a factor of two remains remarkable given the neglect of tropospheric stability (N term), and vertical or horizontal mixing in the 'geostrophic theory'. The lack of vertical mixing (C term) will cause the theoretical treatment to over predict magnitudes. In this way, we put forward the geostrophic theory not as an accurate replacement for running the FFT solver, but rather as an aid to understanding the role of geostrophic adjustment in these complex interactions for which confidence is gained by this comparison with the FFT result. The authors agree that as currently written the above points are not adequately described. To remedy this, the authors propose to revise the manuscript with the following or similar addition at the end of Section 8:**

> **"The geostrophic theory developed in this section provides some useful insight into how the interaction of the inversion layer displacement, pressure field and Coriolis term affects the behaviour in the wake. The comparisons with the full FFT solution show a qualitative match and an order of magnitude quantitative match. The match is sufficiently close, and discrepancies attributable to known simplifications (especially the neglect of vertical mixing which would diminish the geostrophic balance effect) that the comparison adds confidence in the theoretical description as being of value. The geostrophic theory is not here recommended as a replacement for running the full FFT model, but rather as a useful aid to interpreting the role of geostrophic adjustment."**

Similarly, it is hard to evaluate how applicable the results are to real-world wind farm setups. In particular, the choice of model parameters and constants needs clearer justification. Explaining how specific values were chosen would increase confidence in the conclusions. Most importantly, how is the value of C, meaning f/C, determined? There should be a more thorough discussion of what values are realistic for actual wind farms. Implicitly, presenting results for f/C>1 implies these values are possible. Is that actually the case?

**The most commonly used case in this paper if f=C=0.0001 which is representative of mid-latitude, stable conditions. f/C exceeding unity does occur frequently in reality for more stable conditions. f/C being less than unity occurs more frequently for less stable, neutral or unstable conditions. The authors agree that this should be made more clear and more fully explained.**

**The parameter space for wind farms is huge and we cannot look at every "real" case. Instead, we attempted to show trends. Our closed form expressions should allow the reader to examine parameter influence.**

The introduction would benefit from a stronger connection to the existing literature, especially regarding the role of the Coriolis force in wind turbine and wind farm wake recovery, several relevant studies that have not yet been discussed are:
• Dörenkämper et al. (2015), J. Wind Eng. Ind. Aerodyn., 144, 146–153
• Abkar & Porté-Agel (2016), Phys. Rev. Fluids, 1(6), 063701
• Nouri et al. (2020), Applied Energy, 277, 115511
• Englberger et al. (2020), Wind Energy Science, 5, 1359–1374
• Qian et al. (2022), Energy, 239, 121876

**These are valuable references. We will try to include them, but we felt they were less relevant to our work than the papers we cited. The authors also recognise, in the light of overall comments, that additional clarification of the distinction between the present work (focussing on the role of Coriolis forcing via geostrophic adjustment) compared to other work (on other, related aspects of Coriolis contribution to wake behaviour) would be useful in a revised manuscript.**

In both the abstract and introduction, the definitions of the "Rayleigh contribution" and "Coriolis contribution" to wind farm wake recovery need clarification. These terms are not common in wind farm literature, and their meanings only become clearer later in the technical sections. This could confuse readers who are unfamiliar with the specific framework used here. In particular, the "Rayleigh contribution" concept needs more explanation. I especially want to see some justification for the constant C used in this context.

**This is useful feedback. The authors would be happy to more fully explain the Rayleigh friction/vertical mixing terms and Coriolis force term, and to explain and justify the value of C used. A desire to write a brief and compact paper has perhaps gone too far here and sacrificed clarity. A strategy of our analysis was to identify how Turbulence (i.e Rayleigh friction) and Coriolis force contribute to wake recovery. We think this formulation works well at sharpening the questions, so it is important to be clear on what these quantities represent in the formulation used.**

While focusing on these two contributions is helpful (as mentioned on line 52), other effects such as pressure effects and turbulent momentum fluxes, are also important. Including a brief explanation in the introduction that there are different approaches to analyzing the flow would be beneficial.
 **The reviewer may have missed the fact that our Rayleigh friction is a parametrization of turbulent fluxes. We also dig deeply into how pressure gradients work; especially in geostrophic adjustment.**

The finding that Coriolis effects become more significant as wind turbine size and the extent of wind farms increase. The Coriolis force influences both the structure of the atmospheric boundary layer and the velocity deficit of the wind farm wake. As a result, previous studies (such as Gadde and Stevens (2025) JPCS, 1256, 012026 and Kirby & Howland, J. Fluid Mech., 1008, (2025)) showed that Coriolis-induced wake rotation can be clockwise or counterclockwise depending on atmospheric flow conditions. The latter demonstrates that wake deflection depends on the Rossby number. Recent studies showed that the Coriolis force can lead to significant wind farm wake deflection, see e.g.
• Kasper et al., J. Renewable Sustainable Energy, 16, 063302 (2024)
• Kirby & Howland, J. Fluid Mech., 1008, (2025)
It should be noted that the Coriolis force can influence flow dynamics in multiple ways, and different studies address these effects to varying extents. The present manuscript focuses only on the direct Coriolis and Rayleigh contributions, offering a simplified representation of the broader dynamics.
**Yes. We are aware of these contributions but we decided not to treat veering wind in our analysis.**

Overall, the manuscript's conclusion that Coriolis effects deserve more attention in future research appears reasonable. However, as noted in the manuscript, some parameters may have been selected to enhance the observed effects. Therefore, it is important to justify the chosen values clearly. For example, the wind farm size

of 1600 km$^2$ exceeds the dimensions of current real-world farms, although such a size could be plausible in the future. Nonetheless, this assumption should be explicitly acknowledged and discussed. Crucially, as mentioned above, the expected f/C ratio in real situations should be discussed.

**This very large cluster size is justified in terms of aggregations of offshore wind farms existing and planned. This can be more fully explained in the text. Also as mentioned above, the f/C ratios can be more fully explained.**

**The reviewer makes a fair point here. We have provided examples where Coriolis is significant in wind farm wakes, which indicate that its inclusion merits consideration, but we have not shown that Coriolis force is always significant in farm wakes. We have focused on the mechanism by which Coriolis force may act and how stratification may reduce the impact of Coriolis force. We have given a few examples (Table 2) to show how to compute the non-dimensional parameters.**

RC1 Additional points
• The quality of the figures is currently insufficient. For example, Figure 1 uses a smooth color bar, but the data are plotted in discrete contours, making it difficult to interpret. Also it is impossible to see where the zero line is. **This will be remedied with revised versions of the figures.**
• The velocity deficits in the wind farms appear to develop quite slowly. Can the authors discuss this observation and comment on how this should be interpreted in the context of the model **The selection of a fairly low C magnitude causes this, and can be explained more.**
• In Figure 7, it is written that the agreement between FFT and geostrophic theory is good. While both show the same trend, I disagree on calling this a good agreement, as the amplitude obtained from both differs by nearly a factor of 2, which clearly matters considerably. **This comment has been responded to above.**
• Why is the domain shown in Figures 1, 2, and 3 not symmetrically around the wind farm? **This is purely as a presentational preference. This can be made symmetric as preferred.** Particularly, why is the vertical range in the direction in which the wind farm wake is deflected smaller? **Assuming this refers to Figure 3, again this is a presentational preference. The negative and positive ranges can be made the same.**
• Table 3: Using a reduced gravity value g' = 10 m/s^2 seems unrealistically high for atmospheric boundary layer flows, as shown in Table 4. Why is this value used for the analysis? **Indeed this is unrealistically high. Indeed this is an omission to not have explained that this unrealistic value is intended to emphasise the physical effects at play, rather than to represent reality in this specific case. This can be remedied in the text**
• The Coriolis parameter is sometimes given as 0.000124 and other times as 0.0001. Please use a consistent value or clarify the reason for the variation. **The former value is an extreme value for very high latitudes which serves a purpose to demonstrate an extreme case, and the latter is a typical mid-latitude value which is realistic for current real world scenarios. This can be more clearly explained in the text.**
• Line 30: Some of the referenced works concern mountain meteorology, not wind farms, as the current wording implies. These references should be removed, or this should be clarified. **The authors find that it is beneficial to introduce established works from meteorology as the atmospheric perturbation analysis is apt to explain wind farms interacting with the ABL. A statement to this effect can be added to the paper, as removing the references would remove proper attribution of foundational work.**

**Thank you for the constructive feedback.**

---

## Author Comment (AC2)

These author comments are in response to reviewer comments "RC2". The original comments are quoted here in plain typeface for ease of reference, with author comments in bold.

In the manuscript "Coriolis Recovery of Wind Farm Wakes" the authors present a linearized two-layer model of the effects of the Coriolis force on wind farm wake recovery.

General remarks
Wind turbine and wind farm wakes have been studied extensively using numerical models, both engineering models (e.g., Calaf et al. 2010, Porte-Agen et al. 2020) and numerical weather prediction models (Aitken et al 2014, Rosencrans et al. 2024). The authors developed a linearized two-layer for wind farm wake recovery. The model accounts for the wake recovery by the Coriolis force. While recent work by Heck and Howland 2025 showed that the Coriolis force can play some role in the wind turbine wake recovery that effect is relatively small. Considering the length scale of (in particular) offshore wind farms it can be expect that the Coriolis effect on wind farm wake recovery is larger. However, the study presented in the manuscript does not provide a convincing argument for this hypothesis.

**We agree that we have not offered convincing evidence for the importance of Coriolis force overall, for example in an AEP sense. We have provided examples where Coriolis appears to be significant in wind farm wakes, which indicate that its inclusion merits consideration, but we have not shown that Coriolis force is always significant in farm wakes. Our primary goal was to investigate how Coriolis force might act and how it interacts with stratification. Our closed form expressions (e.g. 31,32,45) allow readers to investigate whatever environmental parameters they choose. If the reader expects to see Coriolis playing a strong role in large offshore wind farms, as appears to be implied by the reviewer comment if we interpret correctly, then the present work provides some new insight into conditions when this may or may not be the case, particularly through the mechanism of geostrophic balance and the relation to farm size.**

While the authors presented an elegant mathematical model that, for the most part, can be treated analytically, there are several assumptions that are not well articulated. First, the linearization used in the derivation does not properly account for the effects of atmospheric boundary layer (ABL) stability. Not accounting properly for the ABL stability effects likely exaggerates the impact of Coriolis force on the wake recover. Furthermore, specific application, wind farm wakes, imposes certain constraints on the problem that are not addressed in the manuscript. For example, wind turbine and wind farm wake recovery under convective atmospheric conditions is significantly faster due to energetic convective eddies, i.e. requires shorter distance from a wind turbine or a wind farm than under stably stratified conditions. The effect of Coriolis force and associated wind veering in a convective atmospheric boundary layer (ABL) are negligible and therefore the approach presented in the manuscript is likely not applicable to such cases, however, this was not considered since ABL turbulence was neglected. Furthermore, wind turbine and wind farm wakes depend also on wind speed. Under weak winds wind turbines either do not operate or generate relatively weak wakes. This means that the impact of wind farm wakes is most significant under near neutral to weakly stably stratified conditions. This is also ignored in the manuscript.

**The reviewer may have missed the fact that we have parametrized ABL turbulent stresses as Rayleigh friction. The reviewer seems to think that we neglected turbulent stresses. We show an important competition between turbulent stresses (parameter C) and the Coriolis force (parameter f).**

**The Rayleigh friction coefficient(s) are where this model is sensitive to ABL stability. In this way, the effects that the reviewer mentions – e.g. Coriolis being relatively unimportant in convective conditions due to high rates of turbulent momentum transfer, are captured in the model. In previous work (see for example references Smith 2007 and Gribben and Adams 2023 in the manuscript) it has been explained and is referred to in Section 3.3, that the Rayleigh friction coefficient is sensitive to ABL stability conditions. In other words, unstable air provides a high value for coefficient C. In the present work, a new expression (see 14b) indicates directly that in that case the turbulent contribution to wake recovery (FRR) dominates, and correspondingly FCR is small.**

The authors treat stratification through reduced gravity, giving values between 0.1 and 10, while never providing a definition of the reduced gravity. If we assume that the reduced gravity is commonly defined as: g' = g \delta \theta / \theta_0 (e.g., Jiang, 2014, JAS), where \theta_0 is ~300, and \delta \theta potential temperature difference between the surface and the top of the boundary layer, then a reasonable value of the reduced gravity for conditions relevant for an operating wind farm is between 0 (neutral stratification) and 0.03 (weakly to moderately stable). Notice that the reduced gravity of 0.1 would mean that the potential temperature difference between the surface and the top of the boundary layer is 30 K. Such strong stability of an atmospheric boundary layer is achievable when the winds and therefore shear are weak. Under such conditions wind turbines do not generate power and therefore there are no wakes.

**The authors agree that the reduced gravity term used in this context needs more explanation. It is referred to at the beginning of Section 5.2 but really requires the reader to go to Smith 2010 reference to understand its use here which is probably too much to ask the reader. The manuscript can be updated to improve on this. In this context, it refers to a step change in potential temperature at the inversion. It does not represent a temperature gradient within the ABL as may have been understood from RC2 comments.**

Finally, the treatment of turbulence mixing induced by the presence of a wind farm is very simplistic and does not account for the stronger mixing and momentum entrainment induced by the shear at the top of the wake.

**As mentioned above, the vertical turbulent mixing is embodied as Rayleigh friction. This is certainly a more simple treatment than is used in other more complex models (e.g. RANS CFD, and many others) in the sense that once you have a coefficient value the model is simple and quick to run. It puts a strong emphasis on selection of a Rayleigh coefficient value which represents the conditions, which the authors consider to be a key challenge in model application to real scenarios. A methodology for selecting a surface-layer-stability-sensitive Rayleigh friction coefficient has been worked out (see references) which is promising and requires further validation. In the current context, the value of the Rayleigh friction formulation is in permitting the analysis of friction vs Coriolis contribution to wake recovery.**

Taking all the above into account I do not recommend the manuscript for publication in the present form. The authors should attempt to put their work in proper context of a realistic conditions under which a wind farm operates. An analysis unconstrained by realistic conditions yields unrealistic results and leads to false conclusions.

**We agree that a clearer explanation that the examples presented represent mostly a very large cluster, stable wind case, with one carefully selected extreme case, would benefit interpretation. It may also be useful to point out that the closed form expressions allow the reader to explore scale and atmospheric conditions effect for themselves very easily, i.e. without having to implement or run an FFT solver.**

Specific remark
- Line 109 – It is stated that "The vertical mixing process is difficult to model." This statement should be qualified – it is difficult to model in simple models like the one presented in the manuscript. **Agreed.**
- Line 110 – Two occurrences of the word "may" should be omitted and/or replaced with "is." **Agreed to modify this. The second "may" can be changed to "are sensitive to buoyancy effects" or similar.**
- Line 112 – It is not clear why is Barstad (2016) cited here when the concpets are fundamental textbook concepts. **We can remove this.**
- Equation (9) – The second term on the left-hand-side should be FRR not FRC. **Thank you . This is a typographical error.**
- Line 177 – "Understanding infinitely wide windfarms" is of no real value, since such a wind farm is unrealistic. **Perhaps we should expand on this. The infinitely wide wind farm modelled by Maas(2023) shows a case where there is a seemingly underdamped harmonic wake recovery response which can easily be compared to eqn 13a. That the '1D' formulation (i.e. infinitely wide) of Section 4.1. seemingly matches, in this regard, a very complex flow solution is interesting. As is of course the work elsewhere in the paper which indicates that geostrophic balance will work to inhibit this response in the real world, i.e. with finite width wind farms.**

- Line 274 – Periodic solutions always wrap around from the exit to the entrace of the domain – the question is how the outflow impacts the inflow and the part of the domain that is of specific interest. **Agreed.**
- Line 301 – Instead of "warm" it should be "farm." **Thank you . This is a typographical error.**
- Table 2 – Instead of "Inversion strength" better would be "reduced gravity." **See note above explaining how the inversion strength is represented as a reduced gravity, and that this can be clarified.**
- Table 3 – The values of "reduced gravity" are not relevant for an operating wind farm, a more realistic values should be chosen. **See note above explaining how the inversion strength is represented as a reduced gravity, and that this can be clarified. With this in mind, we believe that the values are realistic.**
- Equations (42) and (43) – instead of "Deficit(y)" a symbol representing deficit should be defined and used. **This can be changed as suggested.**
- Line 489 – However, the model does not include ABL turbulence and its effects, e.g., under convective conditions. This is a serious omission. **As explained above, this is not the case.**
- Line 516 – Symbol "H" should be defined before it is used. **Agreed, thank you for spotting this.**
- Table 4 – It is not clear why would a reduced gravity value be dependent on the fram size (FS). See line 466. **The non-dimensional farm size FS is dependent on the reduced gravity, see line 466. We are selecting realistic values which help explore the FS parameter space.** In particular, the value of 0.1721 is likely unrealistically large for an operating wind farm. **See note above explaining how the inversion strength is represented as a reduced gravity, and that this can be clarified. This value pertains to a potential temperature step change of 5K, which is considered to be a large but not unrealistic value.**
- Line 540 – It is not clear what is meant by "low wind," turbines do not operate below 3 m/s. **We could clarify this. A lower wind gives more time for Coriolis forces to act before dominated by friction forces, but of course only relevant for wind farms above cut in wind speed. We will consider expanding on this use of 'low wind'**

---

## Author Comment (AC3)

General comments on the two reviews.

The main part of this paper (Sections 4-8) describes the fluid dynamics of how the Coriolis force can act to recover the wake and how it interacts with stratification. This material is unique to our paper and fundamental to wake dynamics. Neither reviewer commented on these sections of the paper. Either they did not read it or they found it to be correct and not worthy of comment.

There are several closed form expressions in the paper regarding Coriolis wake recovery. (e.g. 31,32, 38, 45) . These formulae allow the reader to enter their favorite parameter values and see how it impacts certain wake aspects. Neither reviewer mentions or tries to use these formulae.

The main phenological discovery in the paper was the outer wake acceleration which in some cases is concentrated into the "edge jet". We found that symmetric edge jets can be created by a downstream directed Coriolis force acting on the leftward deflected flow. Neither reviewer commented on this idea. Outer wake acceleration is perhaps illustrated on this figure from Finseras et al, (Marine Policy) 2023. In the lower part of this figure we see wakes with accelerated flow on either side. We don't know yet whether the Coriolis force caused these accelerated regions.

[Figure]

On other sections, they had some useful suggestions that we can use to improve the paper. They were interested in the question of how important Coriolis force is in real cases. Although we did discuss this important point, we do not think we can answer that question definitively,   Parameter space is too large to make a global judgement of this type.

---

## Author Response (AR2)

This file consolidates all of the 'Report #1' and 'Report #2' suggestions for revision provided by referees 1 and 2 respectively, as well as the author responses to these suggestions. The referee suggestions are in plain typeface, the author responses are in **bold**, and where the manuscript has been changed accordingly this is indicated in *italics*.

Both referees have clearly reviewed the manuscript in great detail and offer salient and important observations. The authors appreciate the time and dedication required to accomplish this.
* * *
**Report #1**

Reference correction: "Engleberger et al. (2020)" → "Englberger et al. (2020)".
**Thank you.** *Manuscript corrected accordingly at line 48 in Section 1.*

Line 162: Clarify the definition of C in clearer and more accessible language/format.
**We expect that the referee refers to line 142 rather than line 162. A slightly fuller explanation has been given, making it clearer that C is the constant of proportionality for Rayleigh friction.** *Manuscript has been revised accordingly on lines 142 and 143 of Section 3.3.*

Table 2: Provide more explanation or references for the parameter values (e.g., Troposphere Stability, Lateral Diffusivity).
**References have been added.** *Manuscript has been revised accordingly on lines 396 to 399 of Section 6.*

Farm widths statement: The phrase "Typically, within two or three farm widths" seems too strong without sufficient evidence. Were enough cases analyzed to justify "typically"?
**This is a fair point. We have changed "typically" to "for the case studied".** *Manuscript has been revised accordingly on line 620 of Section 10.*

Line 664: Make clearer that "Coriolis and outer wake acceleration" effects can be important in some scenarios, but will not always be important. This motivates the inclusion of extreme cases to illustrate the effects.

**We expect that the referee refers to line 624 rather than line 664. We have addressed this comment by adding to the closing paragraph of the Conclusions section.** *Manuscript has been revised accordingly on line 631 of Section 10.*
* * *
**Report #2**

While the authors addressed all my specific comments their replies to my general comments still leave some of my concerns regarding the limits of applicability of the simplified two-layer approach to a complex problem of relative impact of Coriolis force on wind farm wake recovery open. In the revised manuscript the authors still did not provide sufficient justification for critical parameters of the model presented in the manuscript as relevant to a realistic situation. The problem of the impact of Coriolis force on wind farm wakes is a complex practical problem that can be treated in realistic way using high resolution numerical simulations. When this problem is treated with a simplified approach there is a significant danger of reaching misleading conclusions.

**The authors do not disagree that high resolution numerical simulations are worthwhile. Analysing simplified models, for example as is pursued in this paper, can allow additional insights to be gained on the underlying mechanisms that are important in the flowfield. It is important to understand all of the assumptions going into the model, which is why they are carefully explained.**

For example, the authors conclude that "When FS > 1, the Coriolis Recover (CR) is effective in accelerating air in the "inner" wake. By pushing/pulling the adjacent "outer" wake air leftward, it also creates narrow "edge jets" to the left and right of the jet. In the opposite case of FS <1, the CR in the inner wake is weak as most of the geostrophic adjustment occurs via the PGF rather than flow acceleration." Here, FS is non-dimensional farm size, defined as a ratio of the farm half width and the Rossby Radius of Deformation (RRD) defined as $\sqrt{(g'H)}/f$. The case FS > 1 implies that the farm half width is larger than RRD. The authors use 0.017 ms-2 as the lower value of the reduced gravity. This value corresponds to a 0.5 K inversion, a very weak inversion that most likely occurs only as a transient state. **The authors agree that this represents a weak inversion. (This value appears in Table 4). Our appreciation that this represents a weak but not unrealistic inversion is based on our analysis of radiosonde data. As is explained in the text, we have selected a value that deliberately constructs a lower limit estimate for FS, and we think it is clear that this is not very precise, but nonetheless results in an order of magnitude, or arguably more, difference between realistic values for small and large FS values.**

A more realistic value for the reduced gravity would be between 0.06 ms-2 (corresponding to a weak 2 K inversion) and 0.3 ms-2. **From our own analysis of radiosonde data, there are soundings where the inversion is undetectable so we do think that 0.5 K is a reasonable low value to assign. Also, the referee may have missed that the Table 4 ranges are used in the paper to demonstrate chiefly that the FS value is less than unity, and similarly to indicate an expected range of Froude number, rather than to assert these conditions (e.g. weak inversion case) are frequently occurring.**

Reduced gravity of 10 ms-2 listed in Table 3 is not realistic and the corresponding case should be omitted. **The authors do not agree with this point. It is made very clear (see lines 390 to 395) that this high value is not realistic, and is included to illuminate stability's role. The inclusion of case c is really useful in highlighting the strong role that the PGF plays in the balance of forces: as it is very strong in case c so its effect can be clearly discerned, thus better understood for realistic cases.**

A realistic atmospheric boundary layer height, H, is between 100 m and 3000 m. Since the Coriolis parameter at 45 degrees latitude is f ~ 0.0001 the range of realistic values of RRD is between 24 km and 316 km. Currently, the largest wind farm in the North Sea, Seagreen wind farm, covers 2830 km2 with the corresponding half width of ~27 km. The conditions when FS > 1 occur only when an atmospheric boundary layer is shallow (i.e. stably stratified) and the inversion capping the boundary layer is relatively week. **Agreed (that the FS>1 condition will be at most very infrequently occurring in the context of current wind farms and wind farm clusters).**

Therefore, the speculation that the observed edge jets could be due to the Coriolis effect (including in the additional reply) is not likely correct. **First of all, the FS>1 condition should not be seen as a threshold either side of which the flow regimes suddenly switch. Flow regimes with FS < 1 but approaching FS=1 will have an increasing degree of the FS>1 type of behaviour. Secondly, the augmentation of the observed edge jets in this paper can be clearly attributed to Coriolis (i.e. Coriolis being added to the force mix) because we can contrast with and without Coriolis force results, e.g. Fig1a versus Fig 1b and in Fig 9. Fig 9 in particular clearly shows accelerations outside the wake with f=0, but much greater accelerations with non-zero f.** A simpler explanation is that the jets are resulting from blockage effects. Blockage effects were previously recognized in both observation and simulations (e.g., Hasager et al. 2023, Sanchez Gomez et al. 2023, Schneemann et al. 2025). **Our statements on edge jets may have been misinterpreted as intending to convey that any real-world observations of accelerations are solely attributable to Coriolis, but this is not the case. A separate study would need to be made to ascertain the physical causes of wake-edge acceleration observations e.g. how much is attributable to aggregated turbine blockage and how much may be attributable to the modelled Coriolis-related edge jets. We draw your attention also to the observation that in our models the Coriolis-related accelerations are at the side of the wakes not the wind farms, see Figure 1: "Cases b and c have small regions of negative deficit (i.e. wind speed above ambient) to the left and right of the wake".**

Considering the assumptions made in the development of the simplified, two-layer model, the authors should address in greater detail the limitations of the model. For example, when applying similar two-layer model, the co-author, Smith (2010, cited in the manuscript) stated: "The present model of pressure gradients, gravity

waves and BL response is oversimplified however. Future work should include wind shear and turbulence within the BL. A numerical large eddy simulation (LES) may be required." **The authors mostly disagree with the view that the simplifications and assumptions inherent in the models developed are not clear. However, in case there is any residual risk of misinterpretation, we have added some words highlighting that this is a purposefully simplified model to the beginning of Section 10.** *Manuscript has been revised accordingly on lines 600 and 601 at the start of Section 10.*

**There is tension in the review comments concerning realism versus understanding basic principles, with the thrust of the work concerning the latter and the referee seeming to prefer the former. We suggest that the insights afforded by the simplified models/closed form solutions developed here are useful and it is hard to conceive how the same insights would be manifest from running high fidelity simulations. We have added a closing statement recognising that validation of modelling results is always constructive in case there is any doubt on that point.** *Manuscript has been revised accordingly on lines 635 and 636 at the end of Section 10.*